# Geodiversity Action Plans as a Tool for Developing Sustainable Tourism and Environmental Education

Lucie Kubalíková [1,2], Aleš Bajer [1], Marie Balková [1], Karel Kirchner [2] and Ivo Machar [3,*]

1 Department of Geology and Soil Science, Faculty of Forestry and Wood Technology, Mendel University in Brno, 61300 Brno, Czech Republic; lucie.kubalikova@ugn.cas.cz (L.K.); ales.bajer@mendelu.cz (A.B.); marie.balkova@mendelu.cz (M.B.)
2 Institute of Geonics of the Czech Academy of Sciences, 60200 Brno, Czech Republic; karel.kirchner@ugn.cas.cz
3 Department of Development and Environmental Studies, Faculty of Science, Palacky University Olomouc, 77146 Olomouc, Czech Republic
* Correspondence: ivo.machar@upol.cz

**Abstract:** A complex approach to geodiversity and landscape in order to foster geoconservation and develop geotourism and geoeducation is usually more effective than isolated protection and promotion of geoheritage sites without wider context. A Geodiversity Action Plan (GAP) represents a reasonable tool for how to follow these goals in cooperation with local stakeholders. This specific document is not focused only on an inventory of sites of Earth science interest in an area, but encompasses all geodiversity (geological, geomorphological, soil and hydrological features, processes, systems and relationships). As geoconservation often goes hand in hand with education, sustainable tourism and promotion, the GAP includes practical proposals for management and rational use of the area's geodiversity and geoheritage. This complex approach is needed as it provides a complement to the site-oriented protection or management and, moreover, it can be perceived as coherent with a geoethical approach. The paper presents a case study from Moravian-Slovak border (a central part of Bílé Karpaty/Biele Karpaty Mountains) where the proposal for GAP (including inventory, assessment and management measures) was elaborated together with local authorities, schools and other stakeholders.

**Keywords:** geoconservation; geotourism; local development; environmental education; geosites



## 1. Introduction

Geodiversity Action Plans (GAP) are considered to be an effective tool to facilitate, structure, inform and record action for geodiversity in a given area [1,2]. They provide a mechanism for establishing actions for geoconservation and related issues. They are not focused only on inventory and assessment of selected sites within an area, i.e., geosites, geodiversity sites sensu Brilha [3], but they put geodiversity and geoconservation efforts into the broader frame of the wider landscape [1,2,4]. Thus, the primary reason of elaborating a Geodiversity Action Plan for a particular area is to conceptualise the geodiversity protection and conservation and to enable the development of such activities that may contribute to fulfil these goals (very often geoeducation and geotourism which both have close links to geoconservation) [1,2]. The history of GAPs is quite young, although conceptual documents of how to manage geodiversity (or—more precisely—sites of geological and geomorphological importance) originated much earlier and overlap with legal conservation efforts in general.

The history of geoconservation began with protecting and managing sites [5]; however, in the last decades, a more comprehensive and complex approach that would encompass the relationships between geodiversity, landscape and society has been needed [4]. Also, the need for action in regard to climatic changes and environmental hazards should be reflected in current, holistic approaches to geodiversity and geoconservation [6–8]. This can also

be achieved through the active participation of local stakeholders (including enterprises, schools or local geological societies and other NNOs) on the preparation of a GAP proposal.

Thus, the complex approach to geodiversity and landscape in order to develop geotourism and foster geoconservation is really desirable; however, in some cases, the potential of specific geodiversity sites and geodiversity in general for geotourism and geoeducation development is not recognized [9–12].

In the Czech Republic, every site or area protected by law (Nature Monuments, Nature Reserves, National Nature Monuments, National Nature Reserves, Protected Landscape Areas, National Parks, etc.) has its care plan that is updated every 10 years [13]. However, on the level of general protection of geodiversity, there is the possibility to protect paleontological findings, karst systems and specific mineral deposits. To a limited extent, some geological and geomorphological sites are protected within a category of Important Landscape Element or they can become a part of territorial system of ecological stability [14,15]. Landscape character (which includes both characteristic relief of an area and cultural elements) can be protected within the category of "Natural Park". However, in these cases, a conceptual document is not elaborated or it has a character of recommendations.

This paper is not focused only on the geodiversity conservation, but also on the possibilities of how a GAP can contribute to the development of sustainable forms of tourism and environmental education. The geotourist and geoeducational activities are usually based on the geodiversity inventory and is done through the exploitation of particular sites. That means that landscape elements may become a tourist resource when recognizing its geotourist attractiveness. This is usually done through detailed field work and by identifying sites of geotourist interest, i.e., sites where particular Earth science phenomena are displayed. The degree of suitability for geotourism and geoeducational activities is then done by assessment of the sites by using specific methods within the concept of geosites/geomorphosites [10].

The aim of this paper is to apply the concept of GAP on the border area regarding the limitations, but also the advantages it may bring. In the Czech Republic, a similar initiative has not been developed yet, so this paper can serve as a justification of this approach to bottom-up management and the conservation of geodiversity and geoheritage.

This case study presents an area located on the Moravian-Slovak border in the surroundings of the main ridge of Bílé Karpaty. The area belongs to the Bílé Karpaty Protected Landscape Area (PLA) on the Czech side and Biele Karpaty PLA on the Slovak side, and there are several Nature Monuments and Nature Reserves, but geodiversity and geoheritage of the area are rather overlooked and are not very frequently used for development of sustainable forms of tourism and geoeducational activities. In the care plans, geodiversity and its management is mentioned, but they are elaborated as separate documents. In this case study, geodiversity can be seen as a unifying topic or entity that overcomes political borders, is reflected in the landscape and culture and represents an inseparable part of the local identity and natural and cultural heritage. Based on the detailed fieldwork and by involving local stakeholders, a proposal of a Geodiversity Action Plan is elaborated and specific activities for geotourism, geoeducation and geoconservation are designed.

## 2. Materials and Methods

The first step of any activity related to geoconservation, geotourism and geoeducation in a particular area is represented by a detailed literature and map review and field work. Based on this, geodiversity elements and specific geosites and geodiversity sites are identified and described, including the cultural aspects related to these entities.

Regarding the elaboration of a GAP proposal, the process was divided into several steps and reflected the general recommendations given by Burek and Potter [16]:

1.  Connecting with local stakeholders (especially via Local Action Group and Association of Municipalities): Connecting with local stakeholders was done through the consultation with Local Action Group (LAG) Východní Slovácko on the Czech side of border and Association of Municipalities—Microregion Javorina on the Slovak side.

The LAG includes numerous organizations and institutions of the region, e.g., schools, local enterprises, active citizens, professional associations, municipalities, societies, sport clubs, whereas Association only include municipalities but also has links to the local stakeholders (e.g., enterprises, schools or NGOs). LAG and Association served as connecting points between local residents (or target groups) and Earth science professionals that elaborated a report about the geodiversity of the area. LAG and Association interviewed the stakeholders and discussed their needs and the possibility of spreading the ideas of GAP and introducing the topics of geoheritage, geodiversity and geotourism within the wider public. A key feature in the establishment of a LGAP is the identification of a shared aim and objectives to meet that aim [16].

2. Preparation of a report that describes geodiversity of the study area. The results of detailed fieldwork and a literature review were put into one report which described in detail geological, geomorphological, hydrological, and soil features of the area. Also, the selected sites were chosen as a potential geosites or geodiversity sites suitable especially for sustainable tourism (geotourism) or environmental education. These sites were assessed qualitatively from the Earth-scientific and geotourist point of view by using the set of descriptive criteria. Based on this, some of them were proposed to be included in the most important sites which are suitable for further development. Some basic activities related to geodiversity use were designed. This report served as a basis for a proper GAP proposal.

3. The report was sent to stakeholders. At the meeting, the comments of stakeholders were discussed, and the particular goals of GAP and SWOT analysis were elaborated.

4. The final step was represented by designing and discussing particular activities supporting the understanding and conserving the geodiversity and its rational and sustainable use (geotourism development, geodiversity and geoheritage promotion, environmental education and programs for local schools, guided tours, presentation of the GAP, local geodiversity and geoheritage at various meetings, on web pages of the organizations, on local press, preparation of information panels and leaflets, possibilities of involvement of target groups as volunteers, etc.).

5. The final GAP proposal was submitted to the authorities or representatives of the LAG Východní Slovácko and Association of Municipalities—Microregion Javorina.

The proper implementation can be supported by the approval of authorities (LAG, Association of Municipalities), but it can work (in specific cases) without official approval. Nevertheless, it is always better to approve the document as the presented activities may be supported financially, personally and organizationally.

*2.1. Study Area*

The study area is located on the Moravian-Slovak border (Figure 1). It includes the municipalities belonging to the Local Action Group Východní Slovácko on the Moravian side of the border and Association of Municipalities—Microregion Javorina on the Slovak side. Two associated municipalities were also included (Lopeník in Czechia and Nová Bošáca in Slovakia).

The target region is characterized by a harmonic cultural landscape with a mosaic of meadows, forests and fields. On both sides of the border, the landscape has been used by humans for a long time (agriculture, pasture, quarrying). Thanks to limited accessibility, the intensity of use has never reached high levels, so the typical landscape character has been preserved to the present day. Currently, the study area is included in the Bílé Karpaty Protected Landscape Area (PLA) on the Moravian side of the border and the Biele Karpaty PLA on the Slovak side. There are also several small-scale protected sites (Nature Monuments, Nature Reserves) that assure protection of the natural heritage of the area [13].

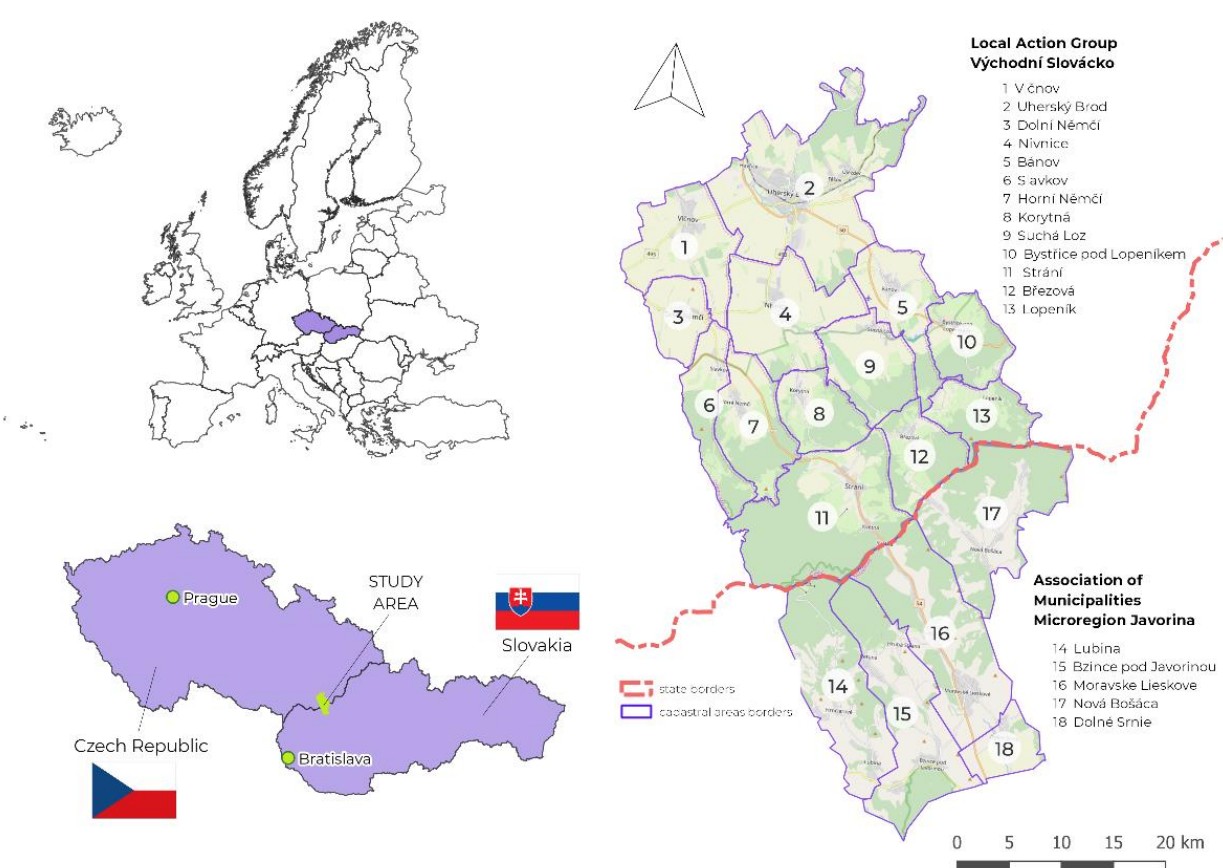

**Figure 1.** Study area and included municipalities (source: OpenStreetMap, available at: https://openstreetmap.cz (accessed on 27 March 2022).

Geodiversity of the Area

Geologically, the study area belongs to the Carpathian System, more precisely to the flysch zone, the Magura mantle group, the Bělokarpatská (White Carpathian) unit. A simplified geological sketch is presented in Figure 2. Flysch is formed by alternating layers of soft sandstones, claystones and siltstones (Figure 3a) which conditions the relief formed by long and flat mountain ridges. The rocks of the Magura group (or White Carpathian units) were deposited in the environment of sea basins and slopes from the Lower Cretaceous. In the Middle and Upper Eocene, sedimentation was terminated [17,18].

During Alpine orogeny, the sedimentary rocks were deformed, folded and pushed to the edge of the Bohemian Massif (geological unit forming a substantial part of the Czech Republic, composed mainly of resistant rocks). This process was accompanied by volcanic activity (so-called neovolcanism-Neogene volcanism), especially along the Nezdenice fault. The pressure and tension created cracks that enabled magma to rise up. However, it did not reach the surface and solidified under it. The volcanic rocks (trachyandezites and trachybasalts) also penetrated into cracks and fissures between the sublayers of claystones, sandstones and siltstones, where it formed veins and lenses. On the contact of volcanic and sedimentary rocks, the contact metamorphism occurred, which resulted in various deformations and transformations of the rocks (e.g., claystone → porcelainite). These phenomena particularly occur on the Moravian part of the study area [18], and examples are given in Figure 3b,c.

On the Slovak side, volcanism did not occur, but numerous limestone outcrops of the Pieniny klippen belt can be found there. The carbonate rocks were deposited during the Mesozoic period. During Alpine orogeny, the blocks of limestone together with the plastic rocks (the already mentioned flysch) were tectonically transported towards the surface.

Due to the different resistance of the rocks, flysch eroded and carbonate rock remained on the surface in the form of remarkable crags and cliffs [19].

The Quaternary is represented by alluvial and slope sediments, river gravels, and sands. Loess and anthropogenic deposits are of limited occurrence.

Geomorphologically, the study area belongs to the Carpathian System, Outer Western Carpathians. Current landscape and landforms are a result of Alpine orogeny and intensive weathering, erosion, transport and accumulation during Quaternary. Due to the orogenic movements, a complex mantle structure originated. Soft rocks were eroded, so the prevailing relief is formed by softly shaped ridges and valleys (Figure 4a). Occasionally, the inversion of relief that resulted from different rock resistance can be observed [20,21]. Volcanic rocks that penetrated through the flysch has also left a trace on the current landscape—they form significant elevations (Hrádek in Bánov, Bučník) thanks to their higher resistance [22]. On the Slovak side, the significant elevations are represented by limestone crags where karst phenomena occur, e.g., the Landrovská cave near Nová Bošáca, karren, small karst cavities filled with secondary calcite and other karst microforms in Bzince quarry [23].

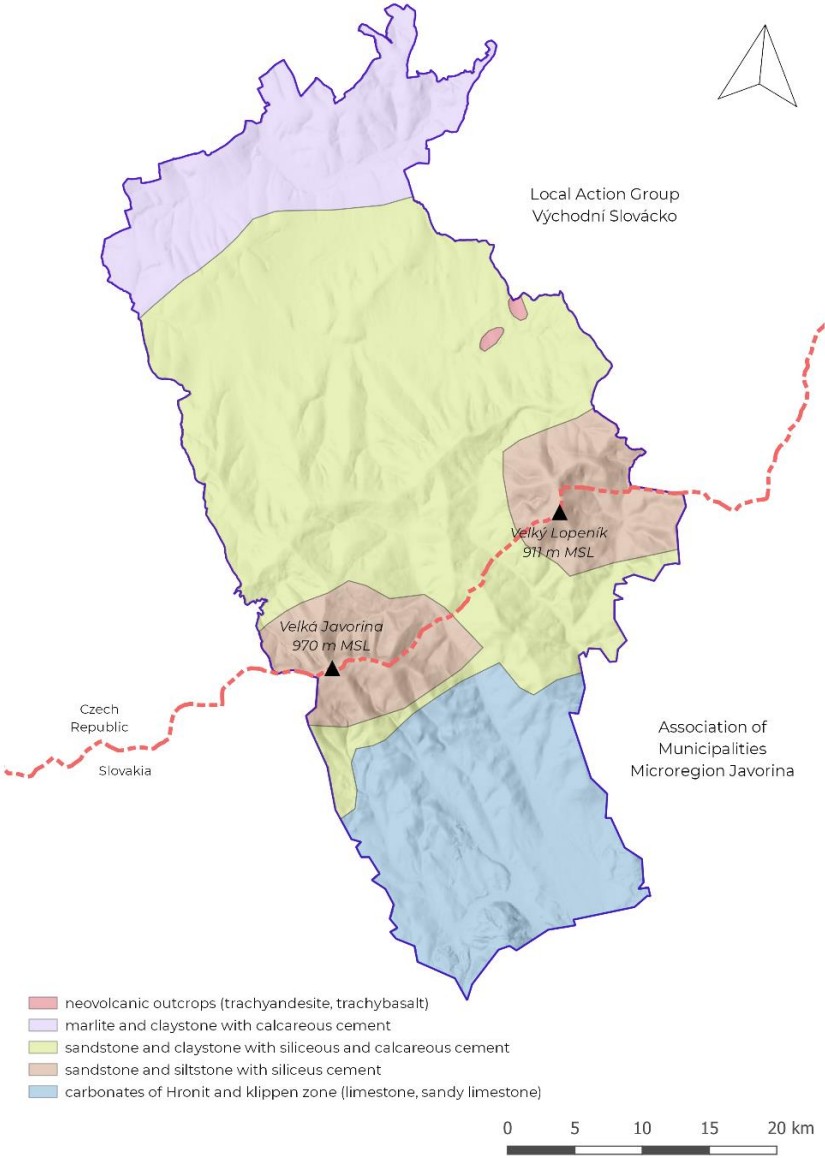

**Figure 2.** Geological sketch of the study area (source: Czech Geological Survey, http://mapy.geology. cz/arcgis/services/Inspire/GM500K/MapServer/WMSServer (accessed on 27 March 2022).

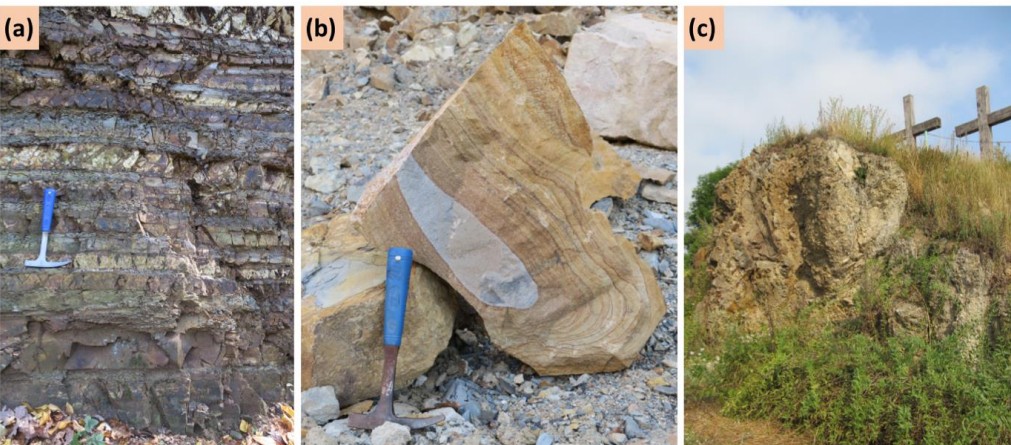

**Figure 3.** (**a**) a detail of flysch at the Skalky Nature Monument; (**b**) porcelanite, Bučník quarry; (**c**) trachyandesite outcrop and remains of old quarry, Hrádek Nature Monument. Photo: authors.

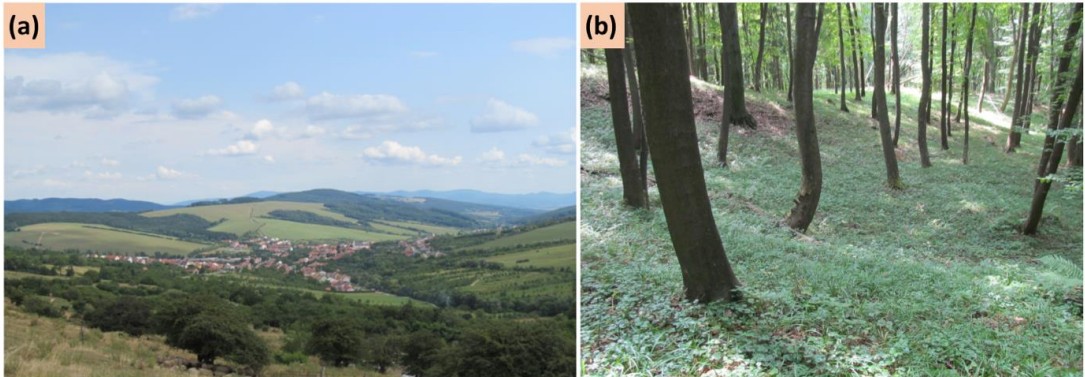

**Figure 4.** (**a**) Strání area, an example of softly shaped ridges and valleys; (**b**) Uvezené Nature Monument, occurrence of landslides is reflected by tree trunks shape. Photo: authors.

The rock composition and character also influence geomorphological processes due to the plasticity of flysch; the landslides and other slope processes are quite common in the study area [24], and an example is shown in Figure 4b.

Anthropogenic landforms are also an important component of the current landscape in the study area, where they are primarily represented by quarries (resistant volcanic rocks, limestone, flysch). At present, old abandoned quarries are often a subject of protection: they are important from an Earth science point of view (as they provide information on the geological development of the area), they have a supporting function for specific habitats and, last but not least, some of them are aesthetically interesting [25].

Soils are represented by chernozems on the lowlands. With increasing altitude, the character of the soils changes, and cambisols, rankers or pararendzinas appear. Occasionally, the gleysols occur, mostly as the result of the presence of practically impermeable claystone layers that allow accumulating subsurface water. Rendzinas can be found on carbonate rocks on the Slovak side.

Hydrologically, the study area is rich in small watercourses. There are several river springs here (Velička, Klanečnice). The water divide between the Morava river and Váh River (both of them belong to Black Sea drainage area) practically corresponds with the Javorina Ridge that also forms a political border between Czechia and Slovakia. The hydrogeological conditions of the area are quite specific (alternation of permeable, less permeable and impermeable layers). They enable the formation of springs and condition landslides and slope movements (Uvezené Nature Monument).

The mineral springs are relatively common in the study area, and they rise up especially along Nezdenice fault in the line Březová–Suchá Loz–Nezdenice–Luhačovice–

Biskupice. Their origin is related to Neogene volcanism and formation of the system of fractures and fissures—the waters in these structures were enriched by carbon dioxide and trace elements and thanks to specific hydrogeological processes, they could reach the surface. The springs of acidulous waters can be found in Březová or Suchá Loz, the springs of hydrogen sulfide waters are situated, for example, in Javorník, Korytná, Strání or Nová Bošáca [26].

Man-made hydrological objects are represented by water reservoirs (e.g., Ordějov, Lubná). Smaller ponds and wetlands are also common. Some flooded quarries are situated close to the border area (e.g., Rasová or Modrá Voda).

Geodiversity has always represented an important resource for human activities in the area. Both volcanic rocks and flysch sediments were quarried and today, some abandoned quarries are legally protected thanks to their scientific value. The material was used for local buildings, e.g., sandstone at Holuby cottage or Lopeník campanile (Figure 5), but also as gravel for roads. Currently, the volcanic rocks (trachyandesites) are quarried at Bučník Quarry, located immediately behind the limit of the study area. Limestones were extracted on the Slovak side of border and used both for lime burning and as construction material. In the Bzince quarry, several types of limestone occur, so it represents a stratigraphic geosite which is very important for geological mapping and research.

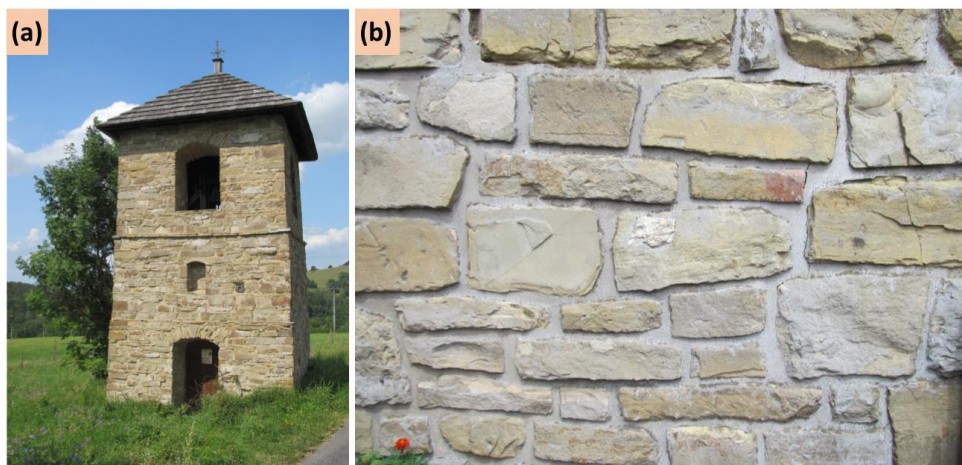

**Figure 5.** (**a**) Lopeník campanille; (**b**) use of local material at Holuby cottage—a detail of sandstones and claystones. Photo: authors.

In the surroundings, a limited occurrence of ores was noted, and old brickyards confirm the existence of several loess and clay pits (e.g., near Vlčnov or Uherský Brod).

Other cultural issues related to geodiversity are represented by toponyms, which often come out from both formal Czech or Slovak languages and local dialects. These local names reflect both landforms (e.g., Žleby—elongated depression in Czech, Bařina—swamp or wet depression in Czech, Loza—type of mineral spring in local dialect, Grúň—hill in Slovak, Čupa—a flat hill in local dialect) and geomorphological processes (Uvezené—in Czech, this word reflects the slope movements; the area where the slope "goes down"). Some toponyms reflect the use of natural resources (Vápenice—in Czech, this word indicates a site where lime was burned).

## 3. Results

Based on the detailed fieldwork and literature review, the geodiversity of the area was described, and 13 geodiversity sites were identified as possible hotspots for geotourist and geoeducation activities. This also enabled the revision of their current legal protection. The analysis and evaluation of geodiversity of the area were done both by professionals and local stakeholders, and it was represented by a SWOT analysis (Table 1). Based on this, particular activities were formulated and a GAP proposal was prepared.

**Table 1.** SWOT analysis as one of the main entries for preparing GAP proposal.

| Strengths | | Weaknesses | |
|---|---|---|---|
| 1. | Harmonic landscape with well-conserved natural heritage | 1. | Geodiversity is not so attractive as the area is rather lithologically and morphologically monotone for the first sight |
| 2. | Geodiversity as a unifying element (similar geological conditions on both sides of the border) | 2. | Some sites are not well accessible, Earth science phenomena are less visible |
| 3. | High added values at specific sites, e.g., ecological, cultural, aesthetical values (the presence of specific ecosystem, protected species, existence of medieval fortresses, historical mining, geodiversity as a part of local identity and heritage) | 3. | A need for the interpretation of Earth-science phenomena if it is going to be used in a suitable way |
| 4. | Present anthropogenic landforms as a bridge between natural and cultural heritage | 4. | Geodiversity is not considered a resource for geotourism |
| 5. | Existing network of marked tourist trails, good "permeability" of landscape, basic tourist infrastructure | 5. | Particular values are not well interconnected (e.g., geodiversity—culture) |
| 6. | Area is not overcrowded by tourists | 6. | Environmental change is endangering some phenomena (e.g., spring areas) |
| 7. | Existing legal protection of geoheritage (or some geological and geomorphological sites) | 7. | Dumping, littering, vandalism at old quarries |
| 8. | Local stakeholders' interest at rational and sustainable use of geodiversity especially for geotourism and geoeducation | 8. | Problems with management of specific sites—accessibility, possibility of using the suitable equipment |
| 9. | Traditional cross-border cooperation | 9. | Lack of finances for geodiversity promotion (e.g., relying on short-term projects and European-funded budgets) |
| Opportunities | | Threats | |
| a. | The area as a calm tourist alternative to more attractive destinations | a. | Inadequate management can lead to degradation or devastation of Earth science phenomena at some sites |
| b. | The promotion of mutual relationships geodiversity—biodiversity—culture can foster the interest in geodiversity—possibility of interconnecting geological and traditional botanical excursions | b. | Lack of interest and finances for promoting and managing geodiversity |
| c. | Knowing geodiversity can strengthen the local identity and belonging to the region | c. | Low interest at geotourism and environmental education |
| d. | Intensifying cross-border cooperation can reinforce and positively influence other issues (not just tourism or education) | d. | Future influence of environmental change |
| e. | Wide range of target group: both visitors to the area and local residents and students | e. | Inadequate use of specific sites (e.g., old quarries as dumps) can become more frequent in the future |
| f. | Rational development of geotourist activities can influence economic growth (e.g., local products and services)—possibility to connect it with a regional trade mark "Slovácko" | | |
| g. | Use of existing geological attractions for education (e.g., geological expositions near watchtowers etc.) | | |

Generally, the final GAP proposal consisted of several parts:

1. Introduction: reasons for elaborating GAP proposal, goals, involved entities
2. A brief introduction about geodiversity and geosites/geodiversity sites
3. SWOT analysis
4. Proposals for the rational use of geodiversity (see later for a more detailed information)
   - Conservation
   - Promotion
   - Education
   - Tourism
5. Conclusions and future prospective, cooperating subjects etc.
6. Attachment 1: A detailed report about geodiversity of the area (result of literature/map reviews and field work, identification, description and qualitative assessment of the sites)
7. Attachment 2: Practical outcomes that can be used and spread (posters, leaflets, educational programs).

Concerning proposals for rational use of geodiversity, the activities have been divided into several groups.

Conservation of natural heritage (or geoheritage) represented the first group. In the report, the revision of conservation measures was done and then discussed with local authorities (municipalities that have a possibility to declare Important Landscape Elements, PLA administration that cares about small-scale protected sites). Based on this, it was decided that the legal protection of the sites of Earth science interest is sufficient and that there is no need for revision, registration or declaration of the new Nature Reserves or Nature Monuments. However, it is necessary to keep these legally protected sites, accept them in landscape or urban planning and ensure that their area is not going to be diminished or that the degree of legal protection won't pass into the lower level. Also, the GAP proposal may help to manage particular sites and keep the Earth-science features visible or comprehensible, e.g., by clearance of self-seeding vegetation, maintaining accessibility of the profiles. This cases are represented primarily by abandoned quarries, e.g., the Skalky Natural Monument, where the visibility of flysch layers penetrated by trachyte vein is limited by trees.

Regarding the promotion of geodiversity, it was found to be insufficient in the study area. Local stakeholders usually did know the importance of geodiversity and its potential for sustainable forms of tourism or environmental education. They also used geodiversity resources and met geodiversity in everyday life, which proved to be a good starting point for the integrated promotion of geodiversity and culture (or geoheritage and cultural heritage). The most effective way of promotion was an implementation of information about geodiversity into the already existing activities and events (e.g., traditional cross-border trips or botanical guided tours). By this, the geodiversity elements were naturally interconnected with biological and cultural features of the landscape and eventually linked to the historical and archaeological aspects. Practically, the promotion was focused on two main directions: the presentation of particular sites where Earth science phenomena can be observed (e.g., rock outcrops, quarries, and viewpoints) and the presentation of general patterns of how geodiversity is reflected in human activities and the use of the landscape (e.g., toponyms that reflect the Earth processes or geodiversity as a resource of construction materials.)

Promotion materials were created to support the idea of geodiversity as a unifying entity and idea of interconnecting geodiversity, biodiversity and culture. Several posters that can be used both for promotion and education were designed: (1) general posters "Geodiversity without borders" (explaining components of geodiversity in the study area), "Geodiversity without borders—a map" (presenting sites of Earth science interest with links to biodiversity and culture), (2) thematic posters: "Volcanoes at your fingertips" (volcanic rocks and landforms), "Water and springs" (focused on hydrography, hydrogeology and

mineral springs), "Limestones" (presenting the limestones of klippen belt including their relationships to cultural heritage). Also, a small leaflet "Geodiversity without borders" was prepared; it included a brief information about Earth-science phenomena of the study area and a timeline of geological development.

All the materials have been elaborated both in the Czech and Slovak language and thus can be used both in the promotion of the area and education in local schools on both sides of the border.

Education represented another group of activities. Besides the abovementioned posters, map and leaflet that may accompany the lectures of geography, biology or history in schools, specific programs for school children were prepared. They were proposed for younger pupils (approximately 8–11 years) and older pupils (12–15 years). Both educational programs consist of two parts: indoor (the presentation of and familiarization with geodiversity of the region, getting practical knowledge via showing the rock samples and discussing specific sites) and outdoor (reading the landscape, rock sampling, exploring the soils of the region, designing proposals for rational use of geodiversity or specific sites such as old quarries). The outdoor part of the school programs is always adjusted to the location of particular educational institution, and, if possible, geodiversity (rocks, landforms, soils) is presented at the local level near the school.

Developing sustainable tourism activities is related to the promotion of the Earth-science phenomena and spreading and presenting these materials on the web pages of the LAG, Microregion, municipalities, NGOs or enterprises that are somehow interested in local development. The coordinators of these activities are the LAG administration on the Czech side and Association administration on the Slovak side of the border. A proposal of a geopath was also discussed with local stakeholders (a path that would connect interesting sites with Earth-science phenomena). A simple geotourist map was prepared within one poster. It included both important geodiversity sites in the study area and geosites situated in its proximity, such as the Rasová sandstone quarry (an example of marine sedimentation) and Haluzická tiesňava (limestone valley with numerous karst features and the demonstration of regressive erosion).

Based on the SWOT analysis and further discussions with local stakeholders both from the Moravian and Slovak side of border, some future goals were specified. They included continuing the cross-border activities (the common promotion of geodiversity as a unifying element—"Geodiversity doesn't know any borders") both based on projects (e.g., Interreg Programe CZ-SK) and local initiatives (led by the LAG and Association of Municipalities). This network is going to be supported also by municipality authorities, schools or local enterprises and NGOs. The Earth-science specialists from universities and research institutions (in this case the Mendel University in Brno and the Institute of Geonics of the Czech Academy of Sciences) in cooperation with local schools will provide educational programs for local students. They will also participate in the monitoring of geosites and geodiversity sites via consultation with Bílé Karpaty/Biele Karpaty PLAs. Concerning other future goals, the stakeholders agreed that volunteering could also be emphasized especially in the management and care about sites of Earth science interest (e.g., clearance of self-seeding vegetation or invasive species obscuring the Earth science phenomena).

## 4. Discussion

GAPs are focused on geotourism and geoeducational issues, which are usually closely related to the geoconservation efforts and can profit from each other [1]. Indeed, geoconservation often goes hand in hand with geotourism development and generally contributes to the sustainability of all the area as proved by the application of the concept to the study area. The role of local communities and local stakeholders (including enterprises, schools or local geological societies and other NNOs) in geoconservation and sustainable geosite management is also vital [27–30]. In this case, the Local Action Group Východní Slovácko and the Association of Municipalities—Microregion Javorina enabled the creation of an

effective network or geodiversity partnership that may serve as a basis for future cooperation and projects. This mutually beneficial relationship can be reflected in a GAP proposal as well.

According to Comer et al. [31], geodiversity is seldom referenced in predominant environmental law and policy. Incorporating geodiversity in planning conservation actions is thus necessary, as geodiversity offers physical support for biodiversity [11,32–34] and provides numerous ecosystem services [4,7,35–38]. In these terms, the implementation of the concepts of abiotic ecosystem services or essential geodiversity variables is desirable for the recognition of the importance of geodiversity [39,40]. These aspects (this kind of evaluation or assessment) can also serve as justification for the elaboration of Geodiversity Action Plans which are focused both on sites of Earth science interest and landscapes and landforms in general [4], no matter the degree of legal protection. In this case, a closer cooperation with authorities responsible for nature conservation is necessary, particularly with regard to monitoring, promotion and future research on geodiversity. The interconnecting geodiversity and biodiversity is also vital for the success of the GAP proposal and its implementation [41], as there is greater emphasis on the wider, non-scientific values of geoconservation including, for example, ecosystem services, links with biodiversity and cultural heritage, geotourism and the benefits for human health and wellbeing through the improved understanding of dynamic landscapes, climate change and natural hazards [42]. This complex approach is needed as it provides a complement to the site-oriented protection and, moreover, it can be perceived as coinciding with a geoethical approach [43].

Concerning the effectiveness of all the processes (elaborating GAP proposal), the overview and evaluation was done two years apart (end of 2021) (Table 2). This procedure was based on Burek and Potter [16].

**Table 2.** Evaluation of elaborating process of GAP proposal and its implementation.

| | |
|---|---|
| Lead organisation(s): | Mendel University in Brno<br>Institute of Geonics of the Czech Academy of Sciences<br>Local Action Group Východní Slovácko |
| Main Partners: | Association of municipalities—Microregion Javorina<br>Municipalities of Lopeník and Nová Bošáca<br>Primary schools in Nivnice, Strání and Moravské Lieskové<br>Bučník quarry operator<br>Representatives of PLAs |
| Funded by: | Interreg CZ—SK operational programme |
| Process: | Described in the chapter Methods; Mendel University and Institute of Geonics as drivers, detailed fieldwork, LAG and Association—local stakeholders, networking, putting local people together |
| LGAP aims: | Promoting geodiversity<br>Recognizing its potential for sustainable forms of tourism and environmental education |
| LGAP objectives: | Mapping and identifying geodiversity elements as well as particular geosites or geodiversity sites<br>Linking geodiversity and culture<br>Involvement of local stakeholders<br>Incorporating geodiversity issues into already existing products and events<br>Proposals for promotion, education and use of geodiversity |
| Strengths: | Close cooperation between local stakeholders and Earth-science professionals<br>Stakeholders interested in topic<br>Practical use of programmes at schools<br>Effective promotion |
| Issues: | GAP proposal not approved by authorities |

Despite the absence of the official approval by local authorities, the process of proposing, elaborating and implementing the GAP can be seen as successful, especially thanks to the close cooperation of Earth science professionals and local stakeholders. There has also been significant openness to the ideas of the GAP that met the expectations of authorities, schools, local NGOs and interest associations, so the implementation of the activities has been quite effective.

Concerning the context of uncertainty and possible negative influences on the implementation of GAP, we did not focus on the economic aspects regarding the influence of geopolitical risks and economic policy uncertainties [44], as this goes beyond the framework of this communication, but in the future, a brief risk assessment of the possible threats regarding implementation may be done by using the simple Risk Assessment Matrix or an extended SWOT analysis [10,45].

**5. Conclusions**

A geodiversity action plan proposal for the central part of Bílé Karpaty/Biele Karpaty Mountains represents an effective tool for managing geodiversity and geoheritage on a local level. In the Czech Republic and in Slovakia, the concept of GAP is not commonly used, but this case study has proven that this methodological approach may work very well on a local level where stakeholders are active and interested (LAG, schools, NGOs, landowners, etc.) and that it represents a solid base for a bottom-up approach to local development and nature conservation.

The GAP was based on a detailed knowledge not only of geodiversity, but also of ecological and cultural aspects of the region. Activities for fostering geoconservation, developing sustainable forms of tourism and promotion of the area are balanced with this knowledge and the needs of the region. It was very enriching and beneficial that local stakeholders who know their region best (municipalities, schools, NGOs, enterprises, active citizens) participated at this proposal and brought their own insights. The application of a multidisciplinary approach when elaborating GAP was also important—integrated promotion may help better understanding of geodiversity and its importance for biotic and cultural components of the landscape, and may justify specific proposals for rational uses of geodiversity (and eventually the finances needed for the management of specific sites).

The main contribution of this case study is that the concept of GAP may be seen as a unifying topic that can serve as a basis for cross-border cooperation and development regardless the different administrative organizations and difficulties with political approval. Practically, this approach may work on the level of Local Action Groups or Association of Municipalities. Furthermore, the practical impact is quite significant—the education and promotion of geodiversity may help realize the possibilities of sustainable tourism development with an extension to geoconservation activities.

In the future, the integration or implementation of a GAP proposal (or geotourism and geoconservation topics) into the strategic planning documents would be desirable. As GAP is based on discussion with and the involvement of the local stakeholders, and it fits well into the Community Led Local Development (CLLD) framework, which represents an effective way of supporting local development projects using structural funds from Europe. Thus, the activities designed in a GAP may be financially and organizationally supported by European funds in the future.

**Author Contributions:** Conceptualization, L.K. and A.B.; methodology, L.K., A.B. and M.B.; data collecting: L.K., K.K. and A.B.; writing—original draft preparation, L.K., A.B., M.B., K.K. and I.M.; writing—review and editing, I.M. and L.K.; visualization, M.B. All authors have read and agreed to the published version of the manuscript.

**Funding:** The research was supported by a project "An Action Plan on the Geodiversity Protection" ("Akční plán pro ochranu geodiversity" in Czech), n. CZ/FMP/6c/01/015 financially supported by European Funds, Interreg V–A Slovak Republic–Czech Republic.

**Institutional Review Board Statement:** Not applicable.

**Informed Consent Statement:** Not applicable.

**Data Availability Statement:** Not applicable.

**Conflicts of Interest:** The authors declare that they have no conflict of interest.

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
