# Peer review of "Geodiversity Action Plans as a Tool for Developing Sustainable Tourism and Environmental Education"

_sustainability, doi:10.3390/su14106043_

Round 1

Reviewer 1 Report

Dear Authors:

The paper analyses an important theme in the context of scientific research on geodiversity and sustainable tourism. The manuscript is taken care of and is in accordance with the fundamental rules for the use of information sources.

Recognizing that your work is an interesting methodological proposal, with special interest to the scientific community, I think it's important: (1) add one more point to the text (between introduction and materials and methods) focused on reviewing literature (state of art) on the theme under study (the importance of geodiversity for the development of new forms of tourism and environmental education, and the role of planning in this process); (2) explain examples of activities of geotourism, conservation and heritage education to be developed in the area of study; (3) identify, in conclusion, the main limitations of the work, as well as the contribution of the study to the advancement of knowledge and its potential for application in other border regions; (4) finally, it is necessary to identify the source of figures 1, 4 and 5.

Author Response

Dear Reviewer, many thanks for your valuable comments to our manuscript. We have corrected the former manuscript following your recommendations (see, please, to attached file with responses of Authors to Reviewers in details). We believe, that all of your comments have been accepted in corrected manuscript. Thank you for your time and effort. Sincerely, Authors

Reviewer 2 Report

I think this paper is quite interesting, but I don't think it's a very good academic paper. This paper is more like a research report or feasibility report. The author spent a lot of time talking about the landscape in Moravian-Slovak cross-border area. However, the author neglected the transformation process from landscape to tourism resources. The author introduced the Geodiversity Action Plan and how to do it. But the author didn't explained why. Thus the paper would be lack of discussion with other related studies. The policy advice also lacked promotional value.

Author Response

(The authors gave the same response as above.)

Reviewer 3 Report

Dear Authors,

I commend you for the effort put into the manuscript on the  Moravian-Slovak cross-border Geodiversity Action Plans. I will make minor comments to be addressed:

  1. The title of the manuscript needs to be concise but still capture the essence of the study.
  2.  The introduction should include the contribution of the study. You mentioned that "Based on the detailed field work and by involving local stakeholders, a proposal of Geodiversity Action Plan is elaborated and specific activities for geotourism, geoeducation and geoconservation are designed" but it does not indicate the benefit of the study to not only the Moravian-Slovak Geodiversity but in general.
  3.  The conclusion needs to be more elaborate, indicating the outcome of the study, the theoretical, and practical implication of the study

Author Response

(The authors gave the same response as above.)

Reviewer 4 Report

Dear Author/s,

Thank you for giving me the opportunity to read your paper. The paper “Geodiversity Action Plans and their contribution to the development of sustainable forms of tourism and nature conservation” is interesting for journal readers. Kindly take note of the following specific comments to make it better.

This study is almost ready to go...I read the paper after revision and also have seems 2 minör revison…

i) #Discussion and conclusions shall be enhanced with the key findings of your study, key messages to be conveyed, usefulness of your outcomes for policy makers and practitioners. In this version, conclusions / policy recommendation are little bit poor. Should make it powerful

ii) Also, need clear future recommendation/implementation in the context of uncertainty.

https://doi.org/10.1177/1354816619888346

https://doi.org/10.1016/j.tourman.2019.06.002

Author Response

See, please, to attached file. Thank you.

Round 2

Reviewer 2 Report

I'm terribly sorry. I'll stick with what I said last time. I don't think there are any significant changes to this article.

Author Response

See, please, attached file. Thank you.

Reviewer 4 Report

Accept